# Health-Related Field-Based Fitness Tests: Normative Values for Italian Primary School Children

**DOI:** 10.3390/jfmk9040190

**Published:** 2024-10-09

**Authors:** Christel Galvani, Fabio Togni, Mariangela Valentina Puci, Matteo Vandoni, Luca Correale, Roberto Codella, Claudio Orizio, Cristina Montomoli, Antonio La Torre, Fabio D’Angelo, Francesco Casolo

**Affiliations:** 1Exercise & Sport Science Laboratory, Department of Psychology, Exercise and Sport Science Degree Course, Università Cattolica del Sacro Cuore, 20162 Milan, Italy; 2Department of Educational Studies, Foreign Literacy and Psychology (FORLILPSI), University of Florence, 50121 Florence, Italy; fabio.togni@unifi.it; 3Unit of Biostatistics and Clinical Epidemiology, Department of Public Health, Experimental and Forensic Medicine, University of Pavia, 27100 Pavia, Italy; mariangela.puci@unipv.it (M.V.P.); cristina.montomoli@unipv.it (C.M.); 4Clinical Epidemiology and Medical Statistics Unit, Department of Medicine, Surgery and Pharmacy, University of Sassari, 07100 Sassari, Italy; 5Laboratory of Adapted Motor Activity (LAMA), Department of Public Health, Experimental Medicine & Forensic Science, University of Pavia, 27100 Pavia, Italy; matteo.vandoni@unipv.it (M.V.); luca.correale@unipv.it (L.C.); 6Department of Biomedical Sciences for Health, Università Degli Studi di Milano, 20133 Milan, Italy; roberto.codella@unimi.it (R.C.); antonio.latorre@unimi.it (A.L.T.); 7Department of Endocrinology, Nutrition and Metabolic Diseases, IRCCS MultiMedica, 20138 Milan, Italy; 8Department of Clinical and Experimental Sciences, University of Brescia, 25123 Brescia, Italy; claudio.orizio@unibs.it; 9Department of Biotechnology and Life Sciences, University of Insubria, 21100 Varese, Italy; fabio.dangelo@uninsubria.it; 10Department of Pedagogy, Exercise and Sport Science Degree Course, Università Cattolica del Sacro Cuore, 20123 Milan, Italy; francesco.casolo@unicatt.it

**Keywords:** health-related physical fitness tests, normative values, physical fitness–motor competence performance index, primary school

## Abstract

**Background/Objectives:** conducting fitness tests in educational settings can lead to advantages for both individuals and groups, beyond just enhancing fitness. With the aim of appropriately interpreting performance, this study was conducted to establish sex- and age-specific percentile normative data for a physical fitness (PF) protocol and to define a compound measure of physical performance. **Methods:** In this cross-sectional study, a short, feasible, and easy-to-administer test battery was performed for 5106 school children, aged 6–10 years. Gender- and age-specific percentiles for the PF tests were constructed, and a Physical Fitness–Motor Competence Performance Index (PF-MC PI) was calculated according to the percentiles. **Results:** Boys scored higher in all the PF tests when compared to girls (*p* < 0.001). There was also a trend towards increased PF levels as the age increased in both genders (*p* < 0.0001). Correlations between scores on individual test items were moderate to high (r from 0.549 to 0.700), and all individual test item scores significantly correlated with the PF-MC PI (*p* < 0.0001). **Conclusions:** the provided percentile values will enable the correct interpretation and monitoring of the fitness status of children and the PF-MC PI can be used for easily rating children’s health-related fitness qualities.

## 1. Introduction

Physical fitness (PF) is nowadays considered to be an important marker of health and the quality of life in childhood. A higher PF level in children has been associated with more positive health-related outcomes, regarding the present and future risks of obesity, cardiovascular disease, skeletal health, and even mental health, in relation to depression, anxiety, mood status, and self-esteem [1,2]. PF in relation to health outcomes and/or health markers is defined as health-related physical fitness (HRPF), and is identified by a variety of factors, including body weight status, cardiorespiratory fitness, musculoskeletal fitness and motor fitness. The importance of monitoring HRPF should be on enhancing physical functional capacity [3]. Motor competence (MC), or motor skill development, is equally important for developing an active and healthy lifestyle in youth [4,5]. Lubans et al. [4] and Cattuzzo et al. [5] also reported a consistent positive association between cardiorespiratory and musculoskeletal fitness and MC and a negative association between MC and body weight across childhood and adolescence. Furthermore, Luz et al. [6] supported the idea that locomotor components (e.g., running and hopping) proved to be the strongest predictor of PF.

Schools are a favorable environment for implementing school practices that support health [7,8]. Schools are also a favorable environment for implementing PF testing, and testing should take place as an integrated aspect of a physical education curriculum [9]. Conversely, fitness testing should not merely remain an isolated part of physical education programs [10]. Teachers reported that they had the desire to assess student fitness [11] by using fitness test results to help students plan for HRPF maintenance or improvement [9]. Fitness tests in educational settings can produce further benefits: assessing children’s health status, identifying children who are at risk of developing certain chronic diseases, tracking children’s fitness performance improvements, increasing children’s physical activity (PA) level, and fostering healthy lifestyle choices. These benefits will be achieved if fitness tests are done in the best interests of youth, with a focus on youth [9,12].

Since the most complete assessments of fitness features require sophisticated laboratory equipment and a high level of tester expertise, they are not always suitable in a school setting. On the other hand, properly conducted field tests offer simple, feasible, and practical protocols, which typically demonstrate good reliability and validity [2,3,13,14]. Twenty-five PF batteries, performed in the school setting with the purpose of monitoring health-related indicators, have been identified. Nine PF batteries were from America (American Alliance for Health, Physical Education, and Recreation—AAHPER; Amateur Athletic Union Test Battery—AAUTB; FitnessGram; National Youth Physical Program—NYPFP; President’s Challenge: Physical Fitness—PCPF; Young Men’s Christian Association Youth Fitness Test—YMCA-YFT; Canadian Association for Health, Physical Education, and Recreation: Fitness Performance Test II—CAHPER-FPT; Canadian Physical Activity, Fitness, and Lifestyle Approach—CPAFLA; Projeto Esporte Brasil—PROESP), nine were from Europe (Eurofit test battery; International Database for Research and Educational Support—INDARES; UNIFITTEST; Physical Fitness Test Battery; SLOfit; FITescola; Adolescents and Surveillance System for the Obesity prevention: Fitness Test Battery—ASSO-FTB; Bouge; Assessing Levels of Physical Activity and Fitness—ALPHA), four were from Asia (Physical Fitness and Athletic Ability Test—PFAAT; Singapore National Physical Fitness Award/Assessment—NAPFA; National Fitness Test Program in the Popular Republic China—NFTP-PRC; Ready for Labor and Defense—GTO), two were from Oceania (Australian Council for Health, Education, and Recreation—ACHPER; New Zealand Fitness Test—NZFT), and one from the Middle East (International Physical Fitness Test—IPFT) [15].

Focusing upon research conducted in Europe, on children aged 6–10 years, we identified that different physical fitness reference standards have been developed in the last few decades [16,17,18,19,20,21,22,23]. Unfortunately, these test batteries take a very long time to be administered and may be ill-suited for testing children during physical education lessons, especially if the tests may be run by generalist teachers. Nevertheless, determining a compound measure of children’s physical performance could be useful for teachers to evaluate children’s overall fitness status. To date, only a few studies proposed the calculation of a total score, but in some cases they did so without specifying percentile values or used methods that may pose challenges for implementation by teachers. Fjørtoft et al. [24] studied a total test score, calculated as the average of z scores for all the test items successfully performed by each child, but this type of calculation is not easy to apply by teachers. In Italy, only one recent study aimed to evaluate the fitness levels in school children from southern Italy, without defining percentile values [25].

To be applicable in a school setting, the test battery should be easy to administer and should not require the specialized training of the experimenters or equipment beyond what is normally available in most gymnasiums. It should be short and include a test battery identifying the principal qualities related to health in children [3]. Furthermore, whereas the previous tests were divided into several age bands with different test items for each age band, the test battery should include the same test items for all ages (6–10 years). Likewise, the compound measure of physical performance should be practical and useful in order to be easily understood by primary school teachers.

Therefore, this study was conducted (1) to establish the normative values of physical fitness and motor competence (PF-MC) components from Italian children using field-based and well-standardized tests that can be easily applied in a school setting and (2) to define a compound measure of physical performance.

## 2. Materials and Methods

### 2.1. Study Design

This is a cross-sectional study. The schools were selected within the context of a project (“A Scuola di sport-Lombardia in gioco”) to which the schools had adhered. Written informed consent, explaining the objectives of the study, was obtained from the parents of all the children. This project was approved by the institutional review board of Regione Lombardia (D.g.r. 9 giugno 2017—n. X/6697) along with the Italian National Olympic Committee (CONI). The study was conducted in accordance with the principles established in the Declaration of Helsinki. The measurements were carried out during one school year, between November and May.

### 2.2. Participants

School children aged 6 to 10 years from 25 elementary schools, belonging to 292 classes, in northern Italy (Lombardy) were recruited for this investigation. PF-MC components were measured for the children who fulfilled the inclusion criteria (having complete data on weight, height, age, and gender). Children with medical issues or physical and mental disabilities were not included in this study. Only the children that performed all the selected tests of the short, field-based PF-MC test battery were included for the analysis. The chronological age with one decimal was calculated for each child as the difference between the test date and the birthdate; the participants were assigned to truncated age categories (e.g., the 7 years category included children aged 7.00–7.99 years) and children exceeding the 10 age category were excluded. Appendix A summarizes the flowchart of the enrollment process of the study sample.

### 2.3. Measurements

All the tests were performed in the respective school gyms during official physical education classes using standardized test protocols. The tests were carried out by scholars with Bachelor or Master of Science degrees in physical activity and education, who had previously completed a 1-day training course to standardize the measurements. A detailed manual of the tests’ instructions was designed and thoroughly read by every physical education teacher before the data collection started.

#### 2.3.1. Anthropometric Measurements

Body weight (kg) and height (cm) were measured for barefoot children, clothed in underwear, using a beam scale (761, SECA GmbH & Co. KG., Hamburg, Germany) with a precision of ±100 g and a portable stadiometer (213, SECA GmbH & Co. KG., Hamburg, Germany) with a precision of ±0.5 cm. The body mass index (BMI) was calculated as body weight (kg) divided by height (m) squared.

#### 2.3.2. Short, Field-Based PF-MC Test Battery

The participants’ PF-MC was assessed by 3 commonly used field-based tests proposed for children. Particularly, the components of the fitness tests were mostly adapted from the ALPHA health-related fitness test battery for children and adolescents [3,14]. Only the proposed 20 m shuttle run test was substituted by the 6 min walking test for a more feasible, safe, and inclusive test [26]. The test battery thus included the following: (1) the 6 min walking test—6MWT—to assess cardiorespiratory fitness; (2) the standing broad jump—SBJ—test to assess musculoskeletal fitness; and (3) the 4 × 10 m shuttle run test—SRT—to assess motor fitness. The SBJ is suitable for the assessment of the hopping locomotor component, and the 4 × 10 m SRT is suitable for the assessment of motor fitness and the running locomotor component. All the tests are reliable and suitable to measure exercise tolerance and endurance thanks to a walking test [27] and to measure MC through the locomotor tests [6]. The time required to conduct and analyze this testing battery for a classroom of 20 children by one physical education teacher was less than 2 h.

The protocols used for fitness testing are described in detail below:

(1) The 6MWT was performed according to the guidelines of the American Thoracic Society (ATS) [28]. The test was conducted in a flat, straight corridor with a hard surface. The children were instructed to walk as fast as possible without running or jogging and were allowed to stop whenever they wanted. Each participant walked continuously for 6 min at a self-selected pace along a 20 m measured tape line, with cones placed at each end of the course. The researchers encouraged the participants with standardized phrases, as described by ATS. No other commands or verbal feedback was given. Specially trained kinesiology graduates, who supervised the test, measured the exact covered distance. The result is given in meters (m).

(2) The SBJ test was performed as described by Castro-Piñero et al. [29]. Briefly, the child stands with his or her feet parallel and shoulder width apart behind a starting line. The child swings the arms backward and forward and jumps with both feet simultaneously as far forward as possible, trying to land with both feet together and maintaining the equilibrium once landed (it was not allowed to put the hands on the floor). The score was obtained by measuring the distance between the last heel mark and the take-off line. Two trials were carried out, and the best result was scored. A further attempt was allowed if the child fell backwards or touched the ground with another part of the body. The result is given in centimeters (cm).

(3) The 4 × 10 m SRT score is the time required to run 10 m 4 times. The procedures were followed as described by Ortega et al. [30]. Briefly, two parallel lines are drawn on the floor (with tape) 10 m apart. At the start line there is one sponge (b), and at the opposite line there are two sponges (a, c). When the start signal is given, the child (without the sponge) runs as fast as possible to the other line and returns to the starting line with the sponge (a), crossing both lines with both feet. The sponge (a) is changed with the sponge (b) at the starting line. Then he/she goes back, running as fast as possible to the opposite line and changes the sponge (b) with the (c) one and runs back to the starting line. Two trials were performed, and the best time was scored. If the child made a procedural error, the performance was interrupted and the test item repeated. The result is given in seconds (s).

A PF-MC Performance Index (PI) for each child and for each test was calculated according to percentiles (percentile P3, P5, P10, score 2; P20, P25, P30, score 4; P40, P50, P60, score 6; P70, P75, P80, score 8; P90, P95, P97 score 10). A cumulative index was then calculated as the average of the PI of the three tests. Appendix A depicts the Physical Fitness–Motor Competence Performance Index (the PF-MC PI) calculation according to the percentiles.

### 2.4. Statistical Analysis

The quantitative variables were summarized as mean values, and the standard deviations and the categorical variables were summarized as percentages. The normality distribution was assessed using the Shapiro–Wilk test, and all the data distributions were checked for outliers (children with recorded data for the 6MWT shorter than 350 m or longer than 850 m, or recorded data for the SBJ shorter than 30 cm or longer than 190 cm, or recorded data for the 4 × 10 m SRT less than 9 s or exceeding 25 s were excluded). The anthropometric and PF-MC test data were grouped by gender and age groups. Differences between the genders and the age groups were analyzed using a 2 (gender: boys, girls) × 5 (age groups: 6, 7, 8, 9, 10 years) multivariate analysis of variance (MANOVA), with the Bonferroni post hoc test.

Percentile charts for 6–10 years were constructed separately for each test by gender using the Learning Management System (LMS) mathematical model [31] implemented in the LMS—chartmaker light software (version 2.54).

Pearson’s correlation coefficients were calculated between the scores on individual test items to study the relationship between the individual test item scores and the PF-MC PI. The strength of the correlation was evaluated according to the following: 0.90 to 1.00 very high, 0.70 to 0.90 high, 0.50 to 0.70 moderate, 0.30 to 0.50 low, and 0.00 to 0.30 negligible correlation [32].

StatView software (5.0.1) was used, except in the LMS method calculations, and the significance level was set at *p* ≤ 0.05.

## 3. Results

There were 5106 participants (mean age 8.6 ± 1.4, 51% boys). The anthropometric data of the study sample, sorted by gender and age groups, are presented in Table 1. Significant main effects of gender were found for body weight (*p* = 0.0048) and height (*p* = 0.0011), with boys having higher values (boys, height: 131.5 ± 9.8 cm, weight: 30.8 ± 8.4 kg, BMI: 17.5 ± 2.9 kg/m^2^; girls, height: 130.9 ± 5.5 cm, weight: 30.2 ± 8.3 kg, BMI: 17.4 ± 2.9 kg/m^2^). Weight, height, and BMI significantly increased with age in boys as well as in girls (*p* < 0.0001 for all ages), with an average increase of 7.2 kg, 12.1 cm, and 1.1 kg/m^2^, respectively.

Overall, all the children were able to perform all of the tests, indicating the suitability of the test battery for children of primary school. Table 2 shows the results of the PF-MC tests by age group and gender. The boys scored higher than the girls in all the field tests: in boys, the mean 6MWT value was 616.7 ± 87.4 m, and in girls it was 607.9 ± 82.6 m (*p* = 0.0002); the boys jumped longer than the girls, with mean SBJ results of 121.6 ± 23.8 cm and 113.7 ± 23.6 cm, respectively (*p* < 0.0001); the boys were faster than the girls, obtaining better results in the 4 × 10 m SRT: 14.5 ± 2.2 s and 15.0 ± 2.2 s, respectively (*p* < 0.0001). Furthermore, the mean scores for all the tests showed increases in both genders across age: the 6MWT values ranged from 529.7 ± 64.0 m for 6-year-old children to 670.3 ± 78.7 m for 10-year-old children (*p* < 0.0001); the SBJ increased from 98.2 ± 18.9 cm at 6 years to 133.8 ± 22.0 cm at 10 years (*p* < 0.0001); the children improved their speed and agility with age, reducing the 4 × 10 m SRT time from 17.1 ± 2.4 s at 6 years to 13.4 ± 1.6 s at 10 years (*p* < 0.0001).

Table 3 displays the respective percentile values (P3, P5, P10, P20, P25, P30, P40, P50, P60, P70, P75, P80, P90, P95, P97) for the 6MWT, SBJ, and 4 × 10 m SRT, stratified by gender and age groups.

Figure 1 depicts the percentile charts of the main components of the PF-MC tests in Italian children, by gender and age group.

The HR-MC PI mean value for the whole sample is 6.4 ± 2.0 (min 2 and max 10). The value remains constant between gender (boys: 6.4 ± 2.0; girls: 6.4 ± 2.0) and between the age groups (6: 6.4 ± 1.9; 7: 6.4 ± 1.9; 8: 6.4 ± 2.0; 9: 6.5 ± 2.0; 10: 6.4 ± 2.0).

The correlations between the scores on the individual test items were moderate to high, with the correlations ranging from 0.549 to 0.700 (*p* < 0.0001). The results indicated that all the individual test item scores significantly correlated with the PF-MC PI (*p* < 0.0001), with the correlations ranging from 0.592 to 0.814 (Table 4).

## 4. Discussion

The aims of the present study were to establish the normative values of PF-MC components from Italian children using field-based and well-standardized tests that could be easily applied in a school setting and to define a compound measure of PF and MC, the Performance Index. Overall, all the children were able to perform all of the tests, indicating the suitability of the test battery for children of primary school age, with boys performing better than girls. With increasing age, the physical fitness scores improved linearly, indicating the adequate sensitivity of the test battery for the age range examined in this study. Age-adjusted percentile curves were obtained for both genders, allowing the creation of the PF-MC PI. Furthermore, all the individual test item scores significantly correlated with PF-MC PI.

Physical fitness was evaluated following a modified ALPHA test battery for youth, which is a valid, reliable, feasible, and safe assessment of health-related physical fitness in this age group [3,13,14]. Muscular strength was assessed using the standing long jump or standing broad jump test, as suggested by the ALPHA test battery. We adopted the SBJ test and excluded the handgrip test due to the number of children to be evaluated. Moreover, the SBJ test might be considered a general index of muscular fitness in youth, and it is practical, time efficient, and low in cost and equipment requirements [29]. The speed–agility component was assessed using the 4 × 10 m shuttle-run test, as suggested by the ALPHA test battery. Cardiorespiratory fitness was assessed through the 6 min walking test, instead of the 20 m shuttle-run test, for a more feasible, safe, and inclusive test. The 6MWT is a reliable and valid functional test for assessing exercise tolerance and endurance in healthy children [27], and extensive research was focused on this test in the last decade [33,34,35], underlying the increase of its usability in this population.

The increase of performance according to age and the differences between male and female school children are in line with other studies in the literature: overall, the older children performed better than the younger children [16,19,23,25], and the boys consistently scored higher than the girls in speed, lower- and upper-limb strength and CRF [17,19,25], with the exception of flexibility, where the girls outperformed the boys [21].

In general, Italian children achieved similar performances to their peers from other countries (Spain, Portugal, and Poland), as far as musculoskeletal fitness and motor fitness are concerned [17,18,19]. In this study, Italian children showed lower cardiorespiratory fitness than other Caucasian counterparts at 6 years of age, for both sexes, but showed higher values at 10 years of age [36]. The slight observed differences may depend on the different statistical methods for centile estimation employed and on possible differences in the measurement protocols applied. The similitude in the PF data may be due to the fact that in all European Union Member States the low levels of health-enhancing physical activity (HEPA) and the high levels of sedentary behavior are alarming and have become a matter of great concern, so much so that most States have developed analogous HEPA policies in the “Sport”, “Health”, and “Education” sectors [37,38].

Proper MC levels are a composite part of children’s health-related fitness, and physical education lessons are suggested to be an important context for developing motor skills [6,39]. MC seems to be relevant to cognitive and social development, aiding in problem-solving, memory, attention, and fostering important social interactions through physical activities [4,40]. Poor motor skills, on the other hand, can contribute to reduced physical activity and negative self-perception, which can affect self-esteem and psychological well-being [5,40]. Due to this, educational and public health policies increasingly emphasize MC as a core aspect of schools’ physical education curricula. Through the promotion of motor skill development in young children, these initiatives aim to prevent future health problems such as obesity and metabolic diseases [41,42].

Many studies have provided percentile values for physical fitness tests for both children and adolescents [16,17,18,19,20,21,22,23,24], evaluating different qualities, such as cardiovascular fitness, musculoskeletal fitness, flexibility, coordination, balance, and motor fitness, though the test batteries were often too long and did not always apply inclusive tests. Our test battery is complete, includes all the HRPF and MC qualities (at least in the locomotor category), and is easy to administer, not requiring special equipment. The tests are suitable for all primary school children, count the same test items for all ages (6–10 years), and can be run by generalist teachers. Besides, the PF-MC Performance Index is an immediate outcome measure of performance.

To the best of our knowledge, our study is the first to provide reference values for gender- and age-specific HRPF and MC for Italian children, including a PF-MC PI. These values may be useful in identifying the children who are at a higher risk for developing unfavorable health outcomes owing to their low fitness level [16], in a school setting, during physical education lessons. Schools are the most suitable settings in which to identify children with poor levels of physical fitness and to promote healthy behavior [36]. The reference values and the PF-MC PI will constitute an important tool in the educational setting, allowing the physical education teachers to start immediate intervention. In order to be effective, PF has to be considered primarily for the purpose of promoting the activity needs of children in the present time and not for the purpose of tracking measures into future years [43], and PF testing should take place in fun, game-like conditions [9].

The psychosocial and educational value of physical activity has been recently highlighted [44], suggesting the teachers’ need for a movement assessment tool that is simple for them to use, quick to administer, and provides valuable feedback to guide future teaching and learning [45]. Indeed, the present work does not aim to produce elements for the simple measurement of motor performance but wants to offer a useful tool for improving the teaching design, favoring the integration of physical activity in the didactic planning for all pupils [46]. In fact, it is important, from a holistic point of view, that participation in sport activities and motor development have significant benefits not only on individual health but also on psychological and social health, as is now reported in the literature [47]. Above all, motor activity in primary school has to improve social attitude in order to realize the inclusion of all pupils, in particular for children who have disabilities or learning-specific disorders. So, it is important for teachers not only to support motor activity but to create a real change in teaching designs, in which motor activity could be the “flywheel” of all classroom activities.

The main limitations of the current study were, first, related to its design (cross-sectional design) and, second, that it included children from only a single region of Italy (Lombardy). Physical fitness reference values in growing children should be preferably obtained from a longitudinal study, which gives the possibility to assess natural changes in individual development. Nevertheless, in the absence of this information, a cross-sectional design analyzed by the appropriate statistical methods seemed to be suitable. Physical fitness reference values should be preferably obtained from a national scale, which gives the possibility to obtain a complete picture of Italian children’s performance. However, the sample size, the inclusion of both rural and urban schools, and the similarity between the fitness values found in our study and those found in a study conducted in the south of Italy [25], all suggest that our data may be representative of Italian children aged 6–10.

## 5. Conclusions

This study has established data on reference values for a short and easy-to-administer (in a school setting) HRPF-MC test battery for Italian, healthy children. Furthermore, this study has defined a useful tool for the measure of performance with an immediate outcome: the PF-MC Performance Index.

Both the percentile values and the Performance Index provided will allow teachers to monitor the PF and MC status of Italian children during physical education lessons, to detect children with low PF and MC levels requiring intervention, and to develop knowledge and behaviors that enable them to acquire or maintain a good, healthy lifestyle.

Furthermore, the performance of the children in the present study was roughly comparable to that of other European children, suggesting the possibility to share HEPA programs across European States.

## Figures and Tables

**Figure 1 jfmk-09-00190-f001:**
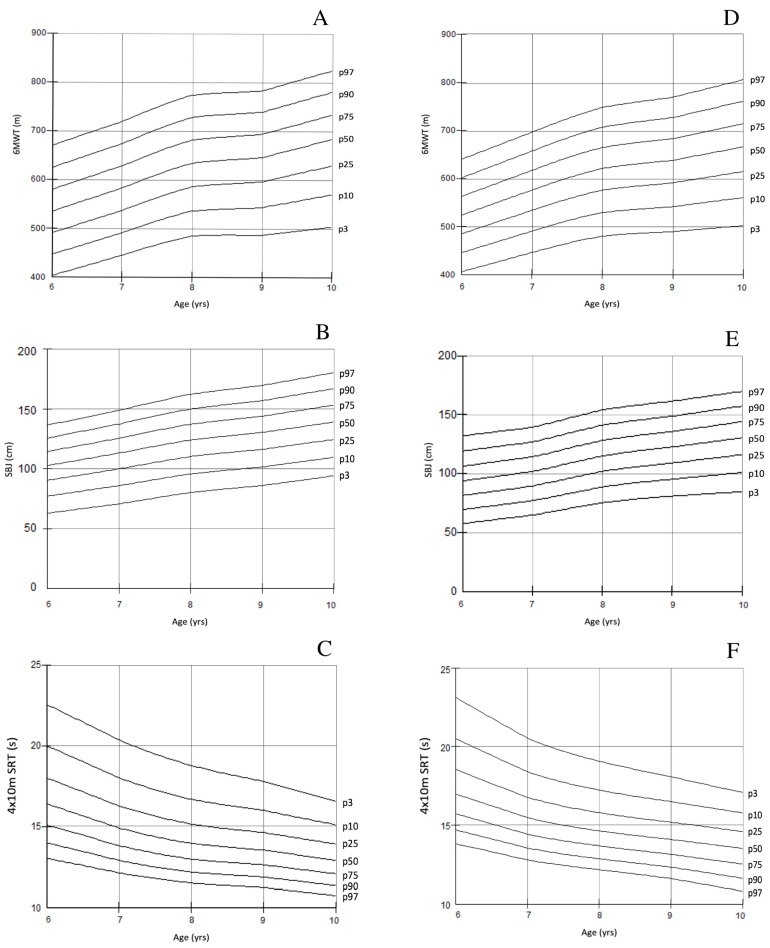
Age-dependent percentiles for the 6MWT (**A**,**D**), SBJ (**B**,**E**), 4 × 10 m SRT (**C**,**F**) in boys and girls.

**Table 1 jfmk-09-00190-t001:** Demographic data stratified by gender and age groups.

Age (yrs)	n	Height (cm)	Weight (kg)	BMI (kg/m^2^)
All children				
6	880	119.1 ± 5.5	23.3 ± 4.4	16.3 ± 2.2
7	1013	125.2 ± 5.7	26.5 ± 5.4	16.8 ± 2.5
8	1079	130.7 ± 6.1	29.7 ± 6.3	17.3 ± 2.8
9	1051	136.0 ± 6.3	33.3 ± 7.2	18.0 ± 2.9
10	1083	142.4 ± 6.8	38.1 ± 8.6	18.6 ± 3.3
Total	5106	131.2 ± 10.1	30.5 ± 8.4	17.4 ± 2.9
Girls				
6	422	118.6 ± 5.4	23.1 ± 4.2	16.4 ± 2.2
7	481	124.5 ± 5.5	26.1 ± 5.1	16.7 ± 2.4
8	536	129.8 ± 5.9	28.8 ± 6.0	17.0 ± 2.9
9	504	135.7 ± 6.5	33.0 ± 7.3	17.8 ± 3.0
10	565	142.4 ± 7.2	37.9 ± 8.4	18.5 ± 3.2
Total	2508	130.9 ± 5.5	30.2 ± 8.3	17.4 ± 2.9
Boys				
6	458	119.6 ± 5.5	23.5 ± 4.6	16.3 ± 2.3
7	532	125.9 ± 5.7	26.9 ± 5.5	16.9 ± 2.6
8	543	131.6 ± 6.1	30.5 ± 6.5	17.5 ± 2.7
9	547	136.3 ± 6.2	33.7 ± 7.1	18.0 ± 2.9
10	518	142.4 ± 6.3	38.3 ± 8.8	18.7 ± 3.4
Total	2598	131.5 ± 9.8	30.8 ± 8.4	17.5 ± 2.9

**Table 2 jfmk-09-00190-t002:** PF-MC tests stratified by gender and age groups.

Age (yrs)	6MWT (m)		SBJ (cm)		4 × 10 m SRT (s)	
All children	Mean ± SD	Min–Max	Mean ± SD	Min–Max	Mean ± SD	Min–Max
6	529.7 ± 64.0	363.0–757.0	98.2 ± 18.9	40.0–171.0	17.1 ± 2.4	11.9–25.0
7	579.0 ± 65.5	366.0–756.0	107.2 ± 19.9	47.0–161.0	15.4 ± 2.0	11.0–23.8
8	626.5 ± 69.2	382.0–820.0	119.3 ± 20.2	60.0–175.0	14.6 ± 1.8	10.8–23.1
9	639.6 ± 72.1	378.0–831.0	126.0 ± 20.8	47.0–187.0	14.0 ± 1.6	10.8–22.2
10	670.3 ± 78.7	400.0–847.0	133.8 ± 22.0	50.0–189.0	13.4 ± 1.6	9.1–19.8
Total	612.4 ± 85.2	363.0–847.0	117.7 ± 24.0	40.0–189.0	14.8 ± 2.3	9.1–25.0
Girls						
6	523.7 ± 59.1	370.0–675.0	94.0 ± 18.9	40.0–171.0	17.4 ± 2.4	12.7–25.0
7	575.1 ± 62.0	379.0–750.0	101.8 ± 18.6	47.0–156.0	15.7 ± 1.9	11.7–22.6
8	619.4 ± 67.1	382.0–820.0	115.0 ± 19.7	65.0–170.0	14.9 ± 1.7	10.9–22.7
9	636.2 ± 68.8	378.0–831.0	122.3 ± 19.7	59.0–184.0	14.2 ± 1.6	11.6–20.3
10	662.6 ± 76.7	420.0–846.0	129.5 ± 21.6	50.0–185.0	13.6 ± 1.6	9.1–19.8
Total	607.9 ± 82.6	370.0–846.0	113.7 ± 23.6	40.0–185.0	15.0 ± 2.2	9.1–25.0
Boys						
6	535.3 ± 67.9	363.0–757.0	102.0 ± 18.2	45.0–150.0	16.8 ± 2.4	11.4–24.5
7	582.5 ± 68.3	366.0–756.0	112.0 ± 19.9	55.0–161.0	15.2 ± 2.0	11.0–23.8
8	633.5 ± 70.6	430.0–820.0	123.6 ± 19.8	60.0–175.0	14.2 ± 1.8	10.8–23.1
9	642.7 ± 75.0	388.0–813.0	129.4 ± 21.3	47.0–187.0	13.8 ± 1.6	10.8–22.2
10	678.7 ± 80.1	400.0–847.0	138.6 ± 21.4	77.0–189.0	13.1 ± 1.5	10.1–18.9
Total	616.7 ± 87.4	363.0–847.0	121.6 ± 23.8	45.0–189.0	14.5 ± 2.2	10.1–24.5

PF-MC: physical fitness and motor competence; 6MWT: 6 min walking test; SBJ: standing broad jump; 4 × 10 m SRT: 4 × 10 m shuttle run test.

**Table 3 jfmk-09-00190-t003:** PF-MC tests norms for Italian children.

6MWT (m)					
Boys Percentile	6 yrs (n = 458)	7 yrs (n = 532)	8 yrs (n = 543)	9 yrs (n = 547)	10 yrs (n = 518)
3rd	403.5	450.0	500.3	503.8	511.6
5th	420.0	468.7	511.0	518.4	537.9
10th	445.9	493.6	540.0	544.0	567.9
20th	479.8	524.0	572.0	578.6	612.8
25th	487.0	534.0	586.0	592.0	625.0
30th	499.0	548.0	600.0	606.0	642.0
40th	518.0	564.0	616.6	622.2	666.6
50th	539.0	580.0	634.0	644.0	685.5
60th	553.0	597.8	653.0	665.0	700.4
70th	571.3	620.0	672.0	687.6	720.0
75th	582.0	630.8	682.0	701.0	735.3
80th	594.0	641.0	693.1	708.4	750.0
90th	624.6	674.7	723.8	740.0	781.2
95th	640.0	700.4	752.8	760.0	803.3
97th	659.2	612.0	760.0	777.8	820.0
Girls Percentile	6 yrs (n = 422)	7 yrs (n = 481)	8 yrs (n = 536)	9 yrs (n = 504)	10 yrs (n = 565)
3rd	417.4	448.9	484.0	506.0	514.0
5th	422.2	467.4	497.1	520.0	528.6
10th	443.0	500.0	533.7	543.5	552.2
20th	472.6	524.4	563.4	576.0	600.0
25th	483.0	537.0	578.0	590.0	610.0
30th	490.9	543.6	586.0	603.0	624.8
40th	510.0	560.8	605.0	620.0	646.0
50th	524.5	575.0	620.0	640.0	666.0
60th	538.8	592.3	637.0	658.0	686.6
70th	556.0	607.7	654.0	675.0	706.0
75th	564.0	616.0	662.0	685.0	716.0
80th	577.4	626.6	677.0	696.0	724.0
90th	602.0	655.6	704.0	722.0	760.0
95th	622.0	678.8	726.0	738.5	790.0
97th	634.3	687.3	742.7	759.6	799.0
SBJ (cm)					
Boys Percentile	6 yrs (n = 458)	7 yrs (n = 532)	8 yrs (n = 543)	9 yrs (n = 547)	10 yrs (n = 518)
3rd	65.8	70.0	86.3	84.4	100.0
5th	71.0	76.0	90.2	92.0	102.0
10th	79.0	85.0	97.0	100.0	109.0
20th	87.0	95.0	106.0	112.1	120.0
25th	90.0	99.0	110.0	116.1	125.0
30th	94.0	102.0	114.0	120.0	128.0
40th	99.0	108.2	119.6	125.0	133.0
50th	102.0	115.0	125.0	130.0	139.0
60th	107.0	119.0	129.0	135.0	144.0
70th	112.0	124.0	134.0	140.0	150.0
75th	114.3	126.0	136.0	143.0	154.0
80th	119.0	130.0	140.0	148.0	158.0
90th	125.0	136.0	150.0	157.0	166.0
95th	130.0	141.4	155.8	163.0	174.1
97th	135.0	145.0	160.0	168.0	180.0
Girls Percentile	6 yrs (n = 422)	7 yrs (n = 481)	8 yrs (n = 536)	9 yrs (n = 504)	10 yrs (n = 565)
3rd	58.7	67.0	76.1	84.3	89.0
5th	64.0	70.0	82.0	89.3	94.0
10th	70.3	78.0	88.7	98.0	101.6
20th	78.6	87.0	98.0	107.0	111.0
25th	82.0	89.0	102.0	109.0	115.0
30th	85.0	93.0	105.0	112.0	119.0
40th	89.0	97.0	110.0	118.0	125.0
50th	94.0	102.0	115.5	122.0	131.0
60th	98.0	106.0	120.0	127.0	136.0
70th	102.1	110.0	126.0	132.0	142.0
75th	105.0	115.0	128.0	136.0	144.0
80th	110.0	117.0	132.0	140.0	148.0
90th	118.0	125.0	140.0	147.5	157.0
95th	126.0	133.0	148.0	153.8	162.7
97th	129.3	137.0	151.0	158.0	169.0
4 × 10 m SRT (s)					
Boys Percentile	6 yrs (n = 458)	7 yrs (n = 532)	8 yrs (n = 543)	9 yrs (n = 547)	10 yrs (n = 518)
3rd	22.2	19.6	18.3	17.4	16.2
5th	21.4	18.7	17.5	16.8	15.8
10th	20.2	17.8	16.6	15.9	15.0
20th	18.7	16.6	15.6	15.0	14.2
25th	18.1	16.2	15.1	14.7	13.9
30th	17.6	15.9	14.9	14.3	13.7
40th	17.0	15.3	14.4	14.0	13.3
50th	16.4	14.8	14.0	13.5	12.9
60th	15.8	14.4	13.6	13.1	12.5
70th	15.3	14.0	13.2	12.8	12.2
75th	15.0	13.8	13.0	12.5	12.1
80th	14.8	13.5	12.7	12.4	11.9
90th	14.0	13.0	12.2	11.9	11.4
95th	13.4	12.6	11.8	11.6	10.9
97th	13.1	12.2	11.7	11.4	10.8
Girls Percentile	6 yrs (n = 422)	7 yrs (n = 481)	8 yrs (n = 536)	9 yrs (n = 504)	10 yrs (n = 565)
3rd	23.4	20.1	18.8	17.7	17.0
5th	22.6	19.6	18.2	17.4	16.4
10th	20.5	18.1	17.1	16.6	15.6
20th	19.2	16.9	16.2	15.5	14.8
25th	18.7	16.7	15.9	15.1	14.5
30th	18.2	16.4	15.5	14.9	14.2
40th	17.6	16.0	15.0	14.4	13.8
50th	17.0	15.4	14.6	13.9	13.5
60th	16.4	14.9	14.2	13.6	13.2
70th	15.9	14.6	13.9	13.3	12.8
75th	15.7	14.4	13.7	13.1	12.6
80th	15.4	14.2	13.4	12.9	12.4
90th	14.7	13.7	12.9	12.4	11.8
95th	14.1	13.2	12.5	12.1	11.3
97th	13.9	12.9	12.3	11.9	10.9

PF-MC: physical fitness and motor competence; 6MWT: 6 min walking test; SBJ: standing broad jump; 4 × 10 m SRT: 4 × 10 m shuttle run test.

**Table 4 jfmk-09-00190-t004:** Pearson correlation coefficients for individual test item scores and PF-MC PI, stratified by gender and age groups.

Test Item	Correlation with PF-MC PI	Correlation with 6MWT	Correlation with SBJ	Correlation with 4 × 10 m SRT
All children				
6MWT (m)	0.595 *	1.00		
SBJ (cm)	0.660 *	0.549 *	1.00	
4 × 10 m SRT (s)	−0.658 *	−0.598 *	−0.700 *	1.00
Girls				
6MWT (m)	0.600 *	1.00		
SBJ (cm)	0.663 *	0.559 *	1.00	
4 × 10 m SRT (s)	−0.663 *	−0.623 *	−0.699 *	1.00
Boys				
6MWT (m)	0.592 *	1.00		
SBJ (cm)	0.675 *	0.541 *	1.00	
4 × 10 m SRT (s)	−0.661 *	−0.574 *	−0.692 *	1.00
6 year				
6MWT (m)	0.726 *	1.00		
SBJ (cm)	0.756 *	0.353 *	1.00	
4 × 10 m SRT (s)	−0.791 *	−0.415 *	−0.557 *	1.00
7 year				
6MWT (m)	0.694 *	1.00		
SBJ (cm)	0.758 *	0.308 *	1.00	
4 × 10 m SRT (s)	−0.800 *	−0.389 *	−0.610 *	1.00
8 year				
6MWT (m)	0.749 *	1.00		
SBJ (cm)	0.766 *	0.396 *	1.00	
4 × 10 m SRT (s)	−0.795 *	−0.483 *	−0.569 *	1.00
9 year				
6MWT (m)	0.707 *	1.00		
SBJ (cm)	0.791 *	0.359 *	1.00	
4 × 10 m SRT (s)	−0.801 *	−0.393 *	−0.627 *	1.00
10 year				
6MWT (m)	0.727 *	1.00		
SBJ (cm)	0.791 *	0.392 *	1.00	
4 × 10 m SRT (s)	−0.814 *	−0.468 *	−0.631 *	1.00

PF-MC PI: Physical Fitness–Motor Competence Performance Index; 6MWT: 6-min walking test; SBJ: standing broad jump; 4 × 10 m SRT: 4 × 10 m shuttle run test; * *p* < 0.0001.

## Data Availability

The data that support the findings of this study will be provided by the corresponding author upon request.

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
