# Peer review of "Health-Related Field-Based Fitness Tests: Normative Values for Italian Primary School Children"

_jfmk, 2024, doi:10.3390/jfmk9040190_

Round 1
Reviewer 1 Report (Previous Reviewer 1)
Comments and Suggestions for Authors
Dear Authors,
For the second time, I have reviewed your manuscript entitled: " Health-related Field-based Fitness Tests: a Test Battery and Normative Values for Italian Primary Schools."
I want to congratulate you on the research you have conducted and for the improvements made, namely:
1) Abstract: The abstract was improved and continued to repeat many keywords that are already present in the title.
2) Introduction: The introduction was improved.
3) Methods Section: Is clear the explanation of the Performance Index and the criteria and calculation of the Physical Performance Index were described in the Method.
4) Discussion: The discussion was improved by referencing some authorities in the field and exploring additional suggestions based on the achieved results.
Once again, I would like to congratulate you on your work.
Best regards
Author Response
Reviewer #1:
Dear Authors,
For the second time, I have reviewed your manuscript entitled: " Health-related Field-based Fitness Tests: a Test Battery and Normative Values for Italian Primary Schools."
I want to congratulate you on the research you have conducted and for the improvements made, namely:
1) Abstract: The abstract was improved and continued to repeat many keywords that are already present in the title.
2) Introduction: The introduction was improved.
3) Methods Section: Is clear the explanation of the Performance Index and the criteria and calculation of the Physical Performance Index were described in the Method.
4) Discussion: The discussion was improved by referencing some authorities in the field and exploring additional suggestions based on the achieved results.
Once again, I would like to congratulate you on your work.
Best regards
We thank the reviewer for her/his appreciation of the revised MS.
Reviewer 2 Report (Previous Reviewer 2)
Comments and Suggestions for Authors
Good job
Author Response
Reviewer #2:
Good job
We thank the reviewer for her/his appreciation of the revised MS.
This manuscript is a resubmission of an earlier submission. The following is a list of the peer review reports and author responses from that submission.
Round 1
Reviewer 1 Report
Comments and Suggestions for Authors
Dear Authors,
I have reviewed your manuscript entitled:" Health-related Field-based Fitness Tests: a Test Battery and Normative Values for Italian Primary Schools." This study was conducted 1) to establish normative values of PF-MC components from Italian children using field-based and well-standardized tests that can be easily applied in a school setting and 2) to define a compound measure of physical performance.
I want to congratulate you on the research you have conducted. It is an interesting study. After completing the revision of the manuscript, I would like to make a few comments:
1) Abstract: The abstract repeats too many keywords already in the title.
2) Introduction: The introduction could be improved.
3) Methods Section: The explanation of the Performance Index is unclear. Although Picture 2 is presented, the criteria and calculation of the Physical Performance Index are not described in the Method.
The statistical analysis was described appropriately and aligned with the study's objectives.
4) Results: For the reader to better understand the acronyms for each variable presented, an explanation of the acronyms should be placed at the bottom of each table.
5) Discussion: The discussion could be improved by referencing some authorities in the field or exploring additional suggestions based on the achieved results. I suggest you look at these articles to improve your discussion.
Once again, I would like to congratulate you on your work. I hope you will consider all my comments.
Best regards
Ramos SA, Massuça LM, Volossovitch A, Ferreira AP and Fragoso I (2021). Morphological andFitness Attributes of Young Male Portuguese Basketball Players: Normative Values According to Chronological Age and Years From Peak Height Velocity. Front. Sports Act. Living 3:629453. doi: 10.3389/fspor.2021.629453
Ramırez-Velez, R, Martinez, M, Correa-Bautista, JE, Lobelo, F, Izquierdo, M, Rodrıguez, F, and Cristi-Montero, C. Normative reference of standing long jump for Colombian schoolchildren aged 9-17.9 years: The FUPRECOL study. J Strength Cond Res 31(8): 2083–2090, 2017.
Garcia-Hermoso, A, Cofre-Bolados, C, Andrade-Schnettler, R, Ceballos-Ceballos, R, Ferna´ndez-Vergara, O, Vegas-Heredia, ED, Ramírez-Vélez, R, and Izquierdo, M. Normative reference values for handgrip strength in Chilean children at 8–12 years old using the empirical distribution and the lambda, mu, and sigma statistical methods. J Strength Cond Res 35(1): 260–266, 2021.
Author Response
Reviewer #1: I want to congratulate you on the research you have conducted. It is an interesting study.
We thank the reviewer for her/his appreciation and for the valuable comments that allow us to better clarify all the manuscript. We agree with the reviewer’s comments and we have carefully revised our manuscript.
R1.1
Abstract: The abstract repeats too many keywords already in the title.
Revised.
R1.2
Introduction: The introduction could be improved.
We have expanded this part of the manuscript.
R1.3
Methods Section: The explanation of the Performance Index is unclear. Although Picture 2 is presented, the criteria and calculation of the Physical Performance Index are not described in the Method.
Amended.
R1.4
Results: For the reader to better understand the acronyms for each variable presented, an explanation of the acronyms should be placed at the bottom of each table.
Amended.
R1.5
Discussion: The discussion could be improved by referencing some authorities in the field or exploring additional suggestions based on the achieved results. I suggest you look at these articles to improve your discussion.
We expanded the discussion and we thank the reviewer for his/her suggestion of considering the three articles.

Reviewer 2 Report
Comments and Suggestions for Authors
Dear authors.
Congrats for the research titled “Health-related Field-based Fitness Tests: a Test Battery and Normative Values for Italian Primary Schools”. It is an interesting research; however, it is necessary to take into account some recommendations so that the manuscript can be improved:
Abstract:
It is recommended to include a brief background to introduce the topic of study.
Introduction:
It is recommended to introduce more information on motor competence, due to the importance of this topic in recent years.
Educational curricula in Europe have slightly changed the perspective of evaluating physical condition towards a broader evaluation, based on other elements. Therefore, it is recommended to better justify this part of the introduction, since the trend is not to use physical tests so much for student evaluations.
Materials and Methods:
The first paragraph of this section should be Study Design.
It is recommended to enter information about the instrument/s used to collect the information in section 2.2.1 Anthropometric measurements
Results:
The results obtained in the study are included, organizedly.
It is recommended to improve the quality of figure 1 since it is very pixelated.
Discussion:
The discussion section should be expanded. For example, physical condition would be convenient to analyze on the one hand, separating the information collected in each test performed. Likewise, something similar should be done with the discussion on motor competence, since there is extensive literature in recent years that allows the results to be compared.
Conslusions:
The aims of the present study were to establish normative values of PF-MC components from Italian children using field-based and well-standardized tests that could be easily applied in a school setting and to define a compound measure of PF and MC, the Performance Index. However, the conclusions do not refer to the normative values of PF-MC components from Italian children. Improve
References:
Ok.
With the application of these changes, the quality of the manuscript will be improved.
Thank you
Author Response
Reviewer #2: Dear authors. Congrats for the research titled “Health-related Field-based Fitness Tests: a Test Battery and Normative Values for Italian Primary Schools”. It is an interesting research.
We thank the reviewer for her/his appreciation. We have carefully revised our manuscript.
R2.1
Abstract: It is recommended to include a brief background to introduce the topic of study.
Revised.
R2.2
Introduction:
It is recommended to introduce more information on motor competence, due to the importance of this topic in recent years. Educational curricula in Europe have slightly changed the perspective of evaluating physical condition towards a broader evaluation, based on other elements. Therefore, it is recommended to better justify this part of the introduction, since the trend is not to use physical tests so much for student evaluations.
We expanded the introduction. We tried to better justify the importance of evaluating physical condition. We stressed the importance of motor competence in the discussion.
R2.3
Materials and Methods:
The first paragraph of this section should be Study Design.
It is recommended to enter information about the instrument/s used to collect the information in section 2.2.1 Anthropometric measurements
Amended.
R2.4
Results:
The results obtained in the study are included, organizedly.
It is recommended to improve the quality of figure 1 since it is very pixelated.
We improved the quality of figure 1.
R2.5
Discussion: The discussion section should be expanded. For example, physical condition would be convenient to analyze on the one hand, separating the information collected in each test performed. Likewise, something similar should be done with the discussion on motor competence, since there is extensive literature in recent years that allows the results to be compared.
We expanded the discussion following reviewer’s suggestions.
R2.6
Conclusions: The aims of the present study were to establish normative values of PF-MC components from Italian children using field-based and well-standardized tests that could be easily applied in a school setting and to define a compound measure of PF and MC, the Performance Index. However, the conclusions do not refer to the normative values of PF-MC components from Italian children. Improve.
Revised.

Reviewer 3 Report
Comments and Suggestions for Authors
The authors did not provide in the Introduction a thorough overview of current physical fitness tests for children and adolescents.
The negative evaluation of the test / tests (?) described on page 2 line 62-74 is unfair and biased. A scientific study requires scientific integrity.
In the Introduction or Methods the authors did not present any credible assumptions/premises for the physical fitness test they developed.
The test proposed by the authors can only be carried out shorter, because the authors’ test consists of a smaller number of trials (exercises) than EUROFIT or the International Physical Fitness Test.
The authors did not prove that conducting fewer tests than those listed in EUROFIT or the International Physical Fitness Test is better or sufficient for assessing the overall physical fitness of students.
Did the authors perform a time calculation, i.e. estimate how much time is required to conduct and analyze the results of individual tests (see: p. 2 line 68)?
No conclusions. The current record is more or less a summary. Please look at the purpose of the research and develop appropriate conclusions. The authors did not present studies related to the test's validity, test's reliability.
Author Response
Reviewer #3:
We thank the reviewer for her/his time. We followed his/her suggestions to improve the manuscript.
R3.1
The authors did not provide in the Introduction a thorough overview of current physical fitness tests for children and adolescents.
Revised.
R3.2
The negative evaluation of the test / tests (?) described on page 2 line 62-74 is unfair and biased. A scientific study requires scientific integrity.
We clarified the paragraph.
R3.3
In the Introduction or Methods the authors did not present any credible assumptions/premises for the physical fitness test they developed.
We modified the Introduction to address this point.
R3.4
The test proposed by the authors can only be carried out shorter, because the authors’ test consists of a smaller number of trials (exercises) than EUROFIT or the International Physical Fitness Test.
The authors did not prove that conducting fewer tests than those listed in EUROFIT or the International Physical Fitness Test is better or sufficient for assessing the overall physical fitness of students.
We changed the title of the paper in order to focus the reader’s attention on the presented normative values instead of the short test battery proposed.
R3.5
Did the authors perform a time calculation, i.e. estimate how much time is required to conduct and analyze the results of individual tests (see: p. 2 line 68)?
We added this information in the Methods.
R3.6
No conclusions. The current record is more or less a summary. Please look at the purpose of the research and develop appropriate conclusions. The authors did not present studies related to the test's validity, test's reliability.
Revised.

Round 2
Reviewer 2 Report
Comments and Suggestions for Authors
Thank you for taking my comments into account.
Author Response
Reviewer #2: Thank you for taking my comments into account.
We thank the reviewer for her/his appreciation of the revised MS.

Reviewer 3 Report
Comments and Suggestions for Authors
Dear Authors,
thank you for you answers.
Your responses were written in a rather unconventional way, e.g. "revised", "we added this information in the Methods". The reviewer usually expects precise, point-by-point answers, not references to an unspecified part of the manuscript. Please provide a precise answer in the "author response". Your current responses do not allow me to change my previous opinion of your manuscript.
Author Response
Reviewer #3:
We thank the reviewer for her/his time. We followed his/her suggestions to improve the manuscript.
R3.1
The authors did not provide in the Introduction a thorough overview of current physical fitness tests for children and adolescents.
The introduction was modified adding the PF tests currently used for children and adolescents:
Twenty-five PF batteries, performed in the school setting with the purpose of monitoring health-related indicators, have been identified. Nine PF batteries were from America (American Alliance for Health, Physical Education, and Recreation_AAHPER, Amateur Athletic Union Test Battery_AAUTB, FitnessGram, National Youth Physical Program_NYPFP, President’s Challenge: Physical Fitness_PCPF, Young Men’s Christian Association Youth Fitness Test_YMCA-YFT, Canadian Association for Health, Physical Education and Recreation-Fitness Performance Test II_CAHPER-FPT, Canadian Physical Activity, Fitness and Lifestyle Approach_CPAFLA, Projeto Esporte Bra-sil_PROESP), nine were from Europe (Eurofit test battery, International Database for Research and Educational Support_INDARES, UNIFITTEST, Physical Fitness Test Battery, SLOfit, FITescola, Adolescents and Surveillance System for the Obesity prevention – Fitness Test Battery_ASSO-FTB, Bouge, Assessing Levels of Physical Activity and Fitness_ALPHA), four were from Asia (Physical Fitness and Athletic Ability Test_PFAAT, Singapore National Physical Fitness Award/Assessment_NAPFA, National Fitness Test Program in the Popular Republic China_NFTP-PRC, Ready for Labor and Defense - GTO), two were from Oceania (Australian Council for Health, Education and Recreation_ACHPER, New Zealand Fitness Test_NZFT), and one from the Middle East (International Physical Fitness Test_IPFT) [15].
R3.2
The negative evaluation of the test / tests (?) described on page 2 line 62-74 is unfair and biased. A scientific study requires scientific integrity.
We clarified the paragraph:
Since the most complete assessments of fitness features requires sophisticated laboratory equipment and a high level of tester expertise, they are not always suitable in a school setting. On the other hand, properly conducted field tests offer simple, feasible, and practical protocols, which typically demonstrate good reliability and validity [2,3,13,14].
R3.3
In the Introduction or Methods the authors did not present any credible assumptions/premises for the physical fitness test they developed.
We modified the Introduction to address this point:
Focusing upon researches conducted in Europe, on children aged 6–10 years, different physical fitness reference standards have been developed in the last decades [16-23]. Unfortunately, these test batteries are very long to be administered and may be ill-suited for testing children during physical education lessons, especially if tests may be run by generalist teachers. Besides, determining a compound measure of children's physical performance could be useful for teachers to evaluate their overall fitness status. To date, only few studies proposed the calculation of a total score, but in some cases without specifying percentile values or using methods that may pose challenges for implementation by teachers. Fjørtoft et al. [24] studied a total test score, calculated as the average of z scores for all test items successfully performed by each child, but this type of calculation is not easy to apply by teachers. In Italy, only one recent study aimed to evaluate the fitness levels in schoolchildren from southern Italy without defining percentile values [25].
R3.4
The test proposed by the authors can only be carried out shorter, because the authors’ test consists of a smaller number of trials (exercises) than EUROFIT or the International Physical Fitness Test.
The authors did not prove that conducting fewer tests than those listed in EUROFIT or the International Physical Fitness Test is better or sufficient for assessing the overall physical fitness of students.
We changed the title of the paper in order to focus the reader’s attention on the presented normative values instead of the short test battery proposed:
Health-related Field-based Fitness Tests: Normative Values for Italian Primary Schools Children
R3.5
Did the authors perform a time calculation, i.e. estimate how much time is required to conduct and analyze the results of individual tests (see: p. 2 line 68)?
We added this information in the Methods:
The time required to conduct and analyze this testing battery to a classroom of 20 children by one physical education teacher is less than 2 hours.
R3.6
No conclusions. The current record is more or less a summary. Please look at the purpose of the research and develop appropriate conclusions. The authors did not present studies related to the test's validity, test's reliability.
We modified the Conclusion:
This study has established data on reference values, for a short and easy to administered (in a school setting) HRPF-MC test battery, in Italian healthy children. Furthermore, this study has defined a useful tool for an immediate outcome measure of performance, the PF-MC Performance Index.
Both the percentile values and the Performance Index provided will allow teachers to monitor PF and MC status of Italian children during physical education lessons, to detect children with low PF and MC levels requiring intervention, and to develop knowledge and behaviors that enable them to acquire or maintain good healthy lifestyle.
Besides, the performance of children in the present study was roughly comparable to that of other European children, suggesting the possibility to share HEPA programs across European States.
Furthermore, according to the suggestions of the two other reviewers:
- In the Introduction we better justified the importance of evaluating physical condition in a school setting
Fitness tests in educational settings can produce further benefits: assess children health status, identify children who are at risk for developing certain chronic diseases, track children fitness performance improvements, increase children physical activity (PA) level, and foster healthy lifestyle choices. These benefits will be achieved if fitness tests are done in the best interests of youth, with a focus on youth [9,12].
- In the Discussion we compared our results with those of other studies
In general, Italian children achieved similar performances than their peers from other countries (Spain, Portugal, Poland) as far as musculoskeletal fitness and motor fitness are concerned [17-19]. In this study, Italian children showed lower cardiorespiratory fitness than other Caucasian counterparts at 6 years of age, for both sexes, but showed higher values at 10 years of age [33]. The slight observed differences may depend on the different statistical methods for centile estimation employed and on possible differences in measurement protocols applied. The similitude in PF data may be due to the fact that in all European Union Member States the low levels of health-enhancing physical activity (HEPA) and the high levels of sedentary behavior are alarming and have become a matter of great concern so that most States developed analogous HEPA policies on “Sport”, “Health”, and “Education” sectors [34,35].
- In the Discussion we stressed the importance of motor competence
Proper MC levels are a composite part of children health-related fitness and physical education lessons are suggested to be an important context for developing motor skills [6,36]. MC seems to be relevant to cognitive and social development, aiding in problem-solving, memory, attention, and fostering important social interactions through physical activities [4,37]. Poor motor skills, on the other hand, can contribute to reduced physical activity and negative self-perception, which can affect self-esteem and psychological well-being [5,37]. Due to this, educational and public health policies increasingly emphasize MC as a core aspect of school physical education. Through the promotion of motor skill development in young children, these initiatives aim to prevent future health problems such as obesity and metabolic diseases [38,39].

Round 3
Reviewer 3 Report
Comments and Suggestions for Authors
Dear Authors,
thank you for your responses. There are unfinished. There is no answer to the question R3.4. Your answers are inconsistent – compare R3.2 and R3.3.
I appreciate you work and a attempt to develop the fitness test for children. Your test is not special (remarkable) and better than other existed test. I even consider that your test is less useful (profitable). Please read about EUROFIT (https://commons.und.edu/cgi/viewcontent.cgi?article=1006&context=ehb-fac)
and IPFT (from 7 year old children) https://www.topendsports.com/testing/international-physical-fitness-test.htm
For me, the biggest limitation of your study is the lack of justification for the selected skill tests. Why were these tests chosen?
In my opinion, in your manuscript you overestimated the value of your test, and underestimated the value other tests.
What is more important, the time of test execution or its reliability, diagnostics, accuracy? What is the value of an invalid test?